# Echo Intensity of Gastrocnemius Is Independently Associated with 6-Minute Walking Distance in Male Patients with Peripheral Arterial Disease

**DOI:** 10.3390/medicina59111894

**Published:** 2023-10-26

**Authors:** Satoshi Yuguchi, Yusuke Ochi, Yukari Sagata, Mitsuhiro Idesako, Shino Maeda, Ryoma Asahi, Masahito Taniguchi

**Affiliations:** 1Department of Physical Therapy, School of Health Sciences, Japan University of Health Sciences, 2-555, Hirasuka, Satte-City 340-0145, Saitama, Japan; r-asahi@jhsu.ac.jp; 2Department of Rehabilitation, Fukuyama Cardiovascular Hospital, 2-39, Midorimachi, Fukuyama-City 720-0804, Hiroshima, Japan; fch_reha@yahoo.co.jp (Y.O.);; 3Department of Cardiology, Fukuyama Cardiovascular Hospital, 2-39, Midorimachi, Fukuyama-City 720-0804, Hiroshima, Japan

**Keywords:** peripheral arterial disease, 6-minute walking distance, gastrocnemius, echo intensity

## Abstract

*Background and Objectives*: This study aimed to examine the differences in the thickness and echo intensity (EI) of the gastrocnemius muscle measured via ultrasonography between healthy adults and patients with peripheral arterial disease (PAD) and to determine the associations of gastrocnemius thickness (GT) and EI within a 6 min walking distance (6MD) in patients with PAD. *Materials and Methods*: This cross-sectional study targeted 35 male patients with PAD (mean age, 73.7 years; mean body mass index [BMI], 23.5 kg/m^2^) and age- and gender-matched 73 male healthy adults (mean age, 73.2 years; mean BMI, 23.3 kg/m^2^). The gastrocnemius thickness (GT) and EI were measured using ultrasound. Both legs of patients with PAD were classified based on higher and lower ankle brachial pressure index (ABI), and the GTs and EIs with higher and lower ABI were compared with those of healthy adults. Multiple regression analysis incorporated 6MD as a dependent variable and each GT and EI with higher and lower ABI, age, and BMI as independent variables. *Results*: This study showed that GT was considerably greater in healthy adults than in both legs with higher and lower ABI (median values, 13.3 vs. 11.3 vs. 10.7, *p* < 0.01), whereas EI was lower in healthy adults than in the lower ABI leg (72.0 vs. 80.8 vs. 83.6, *p* < 0.05). The 6MD was shown to be substantially related to EI in both legs with higher and lower ABIs (*p* < 0.01) but not in the GT. *Conclusions*: In patients with PAD, the GT was lower, and EI was higher than in healthy adults. In addition, EIs in both legs with higher and lower ABIs were independently associated with 6MD in male PAD patients. This study showed that the EI measured via ultrasonography could become an important indicator for treatments for patients with PAD.

## 1. Introduction

The prevalence of peripheral arterial disease (PAD) with intermittent claudication (IC) has recently increased. According to the Trans-Atlantic Inter-Society Consensus II (TASC II), the mortality rate of PAD with IC is 2.5 times higher than that of PAD without IC, and early interventions with exercise therapy are recommended to improve IC [1]. One study found that the metabolic adaptability of skeletal muscle, endothelial function, angiogenesis promotion, and other factors influence the improvement of IC [2], but another study found that the ankle–brachial pressure index (ABI), skeletal muscle metabolism, and endothelial function were not associated with walking distance in PAD [3]. Despite several hypotheses, the factors related to walking distance in PAD remain unknown. Furthermore, abnormalities, such as myopathy and neuropathy, have been found in patients with PAD by muscle biopsy due to ischemia in the lower limbs of the skeletal muscle [4,5]. These abnormalities could limit walking distance in PAD. However, the relationship between the abnormalities and walking distance in patients with PAD presenting IC has not been studied sufficiently. Considering the abovementioned backgrounds, conducting an examination of the association in clinical situations will be difficult because muscle biopsy is invasive.

Recent studies show that ultrasound devices can noninvasively and easily measure skeletal muscle thickness and a percentage of intramuscular fat based on echo intensity (EI) [6,7], which strongly correlates with the rate of intramuscular fat based on the biopsy [7]. Several studies also show that the thickness of quadriceps femoris in people with sarcopenia is lower than in people without sarcopenia, the EI with sarcopenia is higher than without [8], and the EI is related to muscle strength [9] and anerobic threshold [10] in older people. However, there were few studies on evaluating skeletal muscle using ultrasonography, and the relationships of ultrasonography findings with walking distance in patients with PAD presenting IC have been examined insufficiently. Moreover, it has been unclear whether there were differences in ultrasonography findings between healthy adults and patients with PAD. This study hypothesized that the thickness and EI of skeletal muscle measured via ultrasonography would have differences in healthy adults and patients with PAD, and that could be connected to walking distance in patients with PAD. If the hypothesis was ratified, the ultrasonography findings could be used as clinical markers, such as improvements after revascularization and exercise therapy, in patients with PAD.

Thus, the purpose of this study was to clarify the following points: to examine the differences in gastrocnemius thickness (GT) and EI based on ultrasonography between healthy adults and patients with PAD presenting IC and to determine the associations of GT and EI with 6 min walking distance (6MD) in patients with PAD.

## 2. Materials and Methods

### 2.1. Participants

Thirty-five male patients with Fontaine II PAD were recruited, aged 73.7 ± 8.5 years, and scheduled for endovascular treatment between March 2020 and April 2023. None of the patients with PAD required walking aids, had claudication due to stroke or orthopedics, or had unstable general states due to severe heart failure or other disorders. A total of 210 healthy adults aged mean of 74.2 ± 4.3 years, comprising 77 males and 133 females, who participated in medical examinations at Satte City from April 2019 to March 2020 were recruited. The inclusion criteria were based on the following criteria: male participants; those with no drug interventions and medications; diagnosed as having no chronic disease, such as diabetes mellitus, hypertension, hyperlipidemia, chronic kidney disease, cardiovascular disease, stroke, orthopedic disease, and non-sarcopenia by medical practitioners; without a walking aid; and independent in performing activities of daily living. After matching age of healthy adults with that of patients with PAD, finally, 73 male healthy adults were recruited in this study, who were 73.2 ± 4.3 years old, with a mean BMI of 23.2 ± 3.0 kg/m^2^ and were categorized into normal range of 56 (74.0%), and overweight of 17 (23.3%) according to the WHO guideline [11]. This study was performed in accordance with the Helsinki Declaration guidelines, and all participants gave informed consent after explaining our research, such as this study’s purpose and methods, and approved participating in this study. In addition, the aim and protocol of this study were reviewed and approved by the ethics committees of Japan University of Health Sciences (approval number: 3001) and Fukuyama Cardiovascular Hospital (approval number: 59).

### 2.2. Study Design

This is a cross-sectional study. The thickness and EI of gastrocnemius in both legs were assessed using ultrasound in patients with PAD, and 6MD, gait speed, grip strength, and skeletal muscle mass index (SMI) were also measured on the first day before endovascular treatment. In addition, age, body mass index (BMI), history, existence or absence of bilateral lesions, and ABI were obtained from medical records. The thickness and EI of the gastrocnemius in the right leg, as well as age, BMI, gait speed, grip strength, and SMI, were assessed in healthy adults.

#### 2.2.1. Ultrasonography

Using an ultrasound instrument (View’s i; SAKAI Medical Science Co., Ltd., Tokyo, Japan) with a 6 MHz linear array probe, an image of the gastrocnemius muscle was obtained. The measurement of ultrasonography for all participants was conducted using fixed settings, such as B-mode, a fixed dB dynamic range, a fixed gain, and a fixed depth of focus, that were preconfigured for the skeletal muscle and designed to be non-modifiable by the manufacturer. Patients were evaluated while sitting with their knees flexed to 90° and their ankles flexed to 0°. The examiner vertically and gently inserted the probe on the right medial gastrocnemius at the maximum part of the below-knee circumference while monitoring the A-mode on the device screen. The examiner scanned the image of the subcutaneous adipose tissues and gastrocnemius muscles (Figure 1). The subcutaneous fat thickness (SFT) was defined as the distance between the surface and the upper fascia of the gastrocnemius, whereas GT was defined as the distance between the subcutaneous and the deep fascia. Intra-rater reliability was reported in healthy adults when the pressure was <100 gf [12]. The probe pressure on the skin was controlled at 200 gf.

The ultrasound image was converted into Joint Photographic Experts Group files, and the EI of the gastrocnemius was calculated using Adobe Photoshop Elements (Adobe System, Inc., San Jose, CA, USA). The target area was collected, containing as much muscle as feasible without the surrounding fascia, and the area was transformed into an 8 bit grayscale image, with the mean image brightness expressed as a value ranging from 0 to 255, white (Figure 2). The EI of the gastrocnemius was estimated as the mean image brightness [8]. Ultrasonography was assessed in patients with PAD by examiner A and healthy adults by examiner B, who were specialists and well-trained examiners. The repeatability of the GT and EI was assessed by two examiners using the intraclass correlation coefficient (ICC). For examiner A, the ICC for GT and EI was estimated using the 1.2 and 1.1 models, which were >0.9. Similarly, examiner B obtained ICC values of more than 0.96 for GT and EI.

#### 2.2.2. Six-Minute Walking Distance (6MD)

Patients with PAD walked back and forth in the hospital corridor for 30 m. Before 6MD was assessed, patients were told that they should walk as far as they could for 6 min, and if they had significant pain, they could take a break, and once the pain subsided, they could resume walking. In this study, there were no participants who were not able to finish 6MD due to adverse events, such as dyspnea or cardiovascular-related events.

#### 2.2.3. Skeletal Muscle Mass Index (SMI) and Physical Functions

The SMI was assessed in healthy adults using a bioelectrical impedance system device, the MC-780 (TANITA, Corp., Tokyo, Japan), and in patients with PAD using the In Body S10 (IN BODY Japan, Corp., Tokyo, Japan). Healthy adults were measured in the standing posture, whereas patients with PAD were measured in the supine position, and the appendicular skeletal muscle mass (kg) was measured once. SMI was calculated by dividing muscle weight by height squared [8]. Grip strength was measured using a digital dynamometer (T.K.K.5401, Takei Corp., Ishioka, Japan) and defined as the highest value in the right or left hands for each trial. The walking speed (m/s) was measured once and defined as the time it took to walk the 5 m distance at a comfortable pace.

### 2.3. Statistical Analysis

The Shapiro–Wilk test was used for all data to determine normality and before data analysis. Moderate sample sizes were also calculated. The sample size was 14 participants in each group for un-paired *t*-test and X² test (tails, two; α-err, 0.05; power; 0.8). The right and left legs of PAD patients were divided into legs with higher ABI and legs with lower ABI, and SFT, GT, and EI values were compared using the Kruskal–Wallis test between the right legs in 75 healthy adults and legs with higher ABI and lower ABI in 35 patients with PAD. Furthermore, a multiple regression analysis was performed, which included GT and EI in the legs with higher and lower ABI as independent variables, 6MD as a dependent variable, and age and BMI as confounding factors. SPSS ver.27 (IBM Corp., Armonk, NY, USA) was used for statistical analysis, and a significant level of <0.05(two-tailed) was set.

## 3. Results

Table 1 shows the characteristics of 73 healthy adults and 35 patients with PAD. There were no significant differences in age and BMI between the two groups, and the mean values of SMI were 8.0 ± 1.0 in healthy adults and 6.7 ± 0.8 kg/m^2^ in PAD, grip strengths were 36.4 ± 6.0 and 30.1 ± 6.3 kg, and gait speeds were 1.9 ± 0.4 and 0.9 ± 0.2 m/s in patients with PAD (*p* value < 0.01).

Figure 3 shows the actual ultrasound images of healthy adults and patients with PAD. A is an image of a healthy adult aged 73y; subcutaneous fat thickness (SFT) is 2.3 mm; gastrocnemius thickness (GT) is 14.6 mm; echo intensity (EI) is 50.5. B and C are images of a PAD patient aged 75y, and B is the image of the lower limb with higher ABI (0.98); SFT is 3.9 mm, GT is 11.3 mm, and EI is 80.5. C is the image of the lower limb with lower ABI (0.76); SFT is 5.9 mm, GT is 10.7, and EI is 95.9. In Figure 3, the findings were recognized visually that the GT was greater, and the EI was lower in the leg of a healthy adult than in the legs with higher and lower ABI in a patient with PAD.

Table 2 shows the ultrasonography findings in the leg of a healthy adult and each leg with higher and lower ABI in patients with PAD. SFT median values in healthy adults were 2.3 (range: 0.3–9.0) mm and 3.1 (range: 0.5–7.7) mm in the leg with higher ABI and 3.0 (range: 0.7–6.5) mm in the leg with lower ABI, with no significant differences between the three groups. The median values of GT in the leg of healthy adults were 13.3 (range: 8.1–19.5) mm and 11.3 (range: 7.8–15.9) mm in the leg with higher ABI and 10.7 (range: 7.8–18.1) mm in the leg with lower ABI, and both GT in the leg with higher and lower ABI were significantly lower than that in the leg of healthy adults (*p* value < 0.01). The EI media values were 72.0 (range: 37.0–134.7) in the healthy adult’s leg, 80.8 (range: 45.7–140.0) in the higher ABI leg, and 83.6 (range: 43.1–147.0) in the lower ABI leg, with the EI with lower ABI being significantly greater (*p* value < 0.05).

Table 3 shows that the multiple regression analysis that included 6MD as a dependent variable; GT with the higher ABI leg, age, and BMI as independent variables in model 1; EI with the higher ABI leg, age, and BMI as independent variables in model 2; GT with the lower ABI leg, age, BMI as independent variables in model 3; and EI with the lower ABI leg, age, and BMI as independent variables in model 4. GTs with the higher and lower ABI leg had no association with 6MD in models 1 and 3; however, EIs with the higher and lower ABI leg were significantly associated with 6MD in models 2 and 4 (*p* value < 0.01). The powers (1-β) of the multiple regression analysis for 35 participants were 0.81 in model 2 and 0.77 in model 4, which were high values.

## 4. Discussion

This study has shown that GT in patients with PAD was significantly lower and that the EI in the leg with lower ABI in patients with PAD was higher than that in healthy adults. Furthermore, after adjusting for age and BMI, EIs in both legs with higher and lower ABIs were associated with 6MD in patients with PAD. This is the first report that utilizes an ultrasound device to detect skeletal muscle abnormalities in patients with PAD and clarify the relationships between EIs of the gastrocnemius in both the legs and 6MD.

### 4.1. Differences in GT and EI between Healthy Adults and Patients with PAD

Several earlier studies reported on skeletal muscle research in PAD patients. In the result of using computed tomography, the cross-sectional area of the calf skeletal muscle was lower. Intramuscular fat percentage in PAD was higher than in non-PAD after adjusting for age, gender, and BMI. Moreover, the lower the ABI, the higher their differences [13]. The percentage of intramuscular fat in myopathy or neuropathy in neuromuscular disorder increased, which was detected as an increase in EI via ultrasonography [14]. Skeletal muscle degeneration related to ischemia in lower limbs occurred in patients with PAD [15,16]. In this study, healthy adults were not diagnosed with sarcopenia based on the Asia Working Group for Sarcopenia 2019 [17] and had normal skeletal muscle mass and physical functions. Furthermore, there were no variations in age and BMI between healthy adults and patients with PAD. Age, gender, and BMI have been reported to alter the thickness and EI of skeletal muscle evaluated via ultrasonography [6,8]. However, the effects of confounding factors on the GT and EI are excluded in this work. As a result, muscle degeneration and increased intramuscular rates occurred in patients with PAD; as previously reported, ultrasonography could detect lower GT and greater EI than healthy adults in this study. Meanwhile, there was a tendency for lower GT and greater EI in the leg with lower ABI than in the leg with higher ABI, but these differences were not statistically significant. According to reports, in the case of only an ABI of <0.7 in either of the legs or a gap of ABI of >0.2 in both legs, there was a decrease in the cross-sectional area of skeletal muscle and an increase in the intramuscular fat rate in the leg with lower ABI [13]. In our study, there were 17 patients with ABI of <0.7 or a gap of >0.2 in both legs (48.6%), and the values of GT with lower and higher ABI were 11.3 ± 2.2 and 11.4 ± 1.72 mm, respectively; there were no significant differences in both legs. Future studies would be required to determine whether there was a difference in the thickness of skeletal muscle and EI between legs with higher and lower ABI.

### 4.2. Associations of 6MD with GT and EI in Patients with PAD

The EIs in the legs with higher and lower ABIs were independently associated with 6MD, unlike GTs in both legs. Previous studies have reported the correlations of skeletal muscles’ EI with physical functions. One such study, in community-dwelling older people, mentions that the EI of the quadriceps femoris is strongly related to gait speed, as a physical function, after comparing the muscle thickness [8]. In a study using computed tomography for patients with PAD, the percentage of intramuscular fat in the legs with only higher ABI was significantly correlated to 6MD; however, the leg with the lower ABI would, possibly, compensate for that with lower ABI in 6MD [13]. As a result, it is reasonable to conclude that the EI showing intramuscular fat rate was connected to 6MD as a physical function in this study. However, it is debatable whether not only the leg with higher ABI but also the leg with lower ABI were independently associated with 6MD. In terms of the occurrence of bilateral or unilateral lesions, patients with PAD in our study may differ from those in the previous study. In an earlier study, the average values of legs with lower ABI were 0.62, which was the same as in our study, whereas the values in the legs with higher ABI and the rates of the presence of bilateral lesions were not shown. In our study, the rate of the occurrence of bilateral lesions was 31.3% (*n* = 11), and bilateral or unilateral leg pain occurred in patients with bilateral lesions; however, it was unknown which leg pain occurred in 6MD with higher and lower ABIs. Future research is required to demonstrate the above-mentioned point. Meanwhile, it is worth noting that both legs with higher and lower ABIs were independently associated with 6MD, suggesting that it is important for both legs in patients with PAD to undergo exercise therapy to improve 6MD regardless of the ABI values. According to a 2-year longitudinal research, increased intramuscular fat is associated with decreased physical activity in patients with PAD [18].

Important points of this study were that ultrasonography was a noninvasive method that first was able to detect gastrocnemius degeneration due to lower limb ischemia in patients with PAD and, in addition, clarified the association between 6MD and the EI in both legs. These results could contribute to treatments for patients with PAD as an important indicator for not only the effects of revascularization and exercise therapy but also physical activities. In the future, associations between EI measured via ultrasonography as a clinical indicator and improvements in physical performances, such as walking distance, gait speed, and muscle strength, and activities of daily living or occurrence of any adverse events, should be determined.

### 4.3. Limitation

This study has several limitations. First, since this study recruited only male healthy adults and patients with PAD, these results cannot be extrapolated to participants in women. Thus, this study cannot conclusively demonstrate the same results for healthy adults and patients with PAD in women. Second, because health status data, including blood pressure, activity level, and the type of drugs were not collected in healthy adults, the detailed state of participants was unclear. Third, patients with PAD include those with a history of lumber spinal stenosis, which might affect the 6MD values. Fourth, the settings of this ultrasound device were not disclosed because it was patented by SAKAI Medical Science Co., Ltd., and the low EI values obtained in this study could vary when using other ultrasound devices. Finally, this study is a cross-sectional study; it is unknown whether the change of the EI using ultrasonography affects that of 6MD, which should be determined in a longitudinal study in the future.

## 5. Conclusions

Because patients with PAD had lower GT and higher EI than age- and gender-matched healthy adults, ultrasonography could detect muscle abnormalities, such as muscle atrophy and the increase in intramuscular fat in patients with PAD. Additionally, adjusting for age and BMI, the EI in the legs with higher and lower ABIs showed an independent association with 6MD in patients with PAD presenting IC. This study showed that the EI obtained via ultrasonography could become an important indicator to detect the effects of interventions on the lower limb skeletal muscle, which could contribute to treatments for male patients with PAD presenting IC.

## Figures and Tables

**Figure 1 medicina-59-01894-f001:**
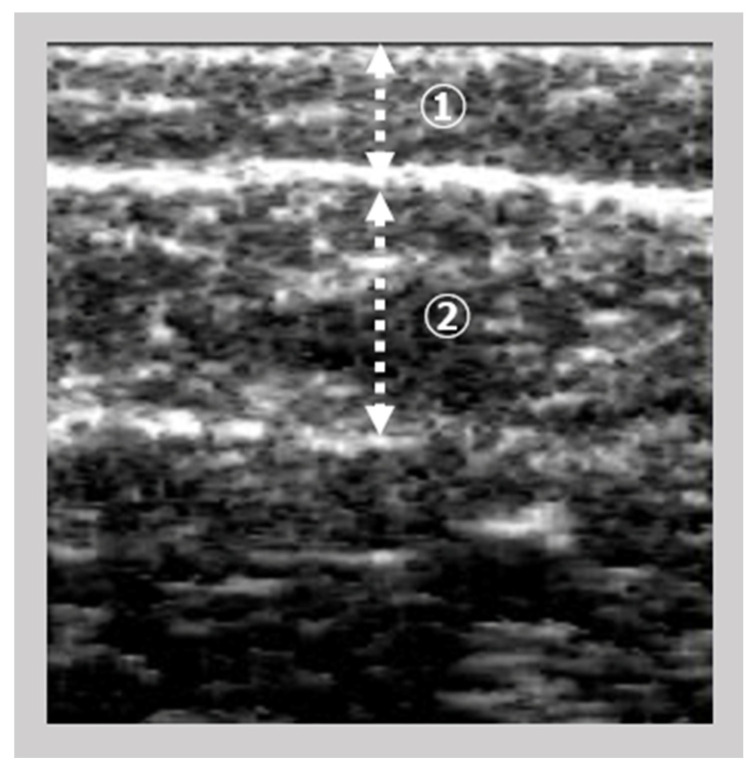
The image of subcutaneous fat and gastrocnemius thickness by ultrasonography. ➀ Subcutaneous fat thickness; ② Gastrocnemius thickness.

**Figure 2 medicina-59-01894-f002:**
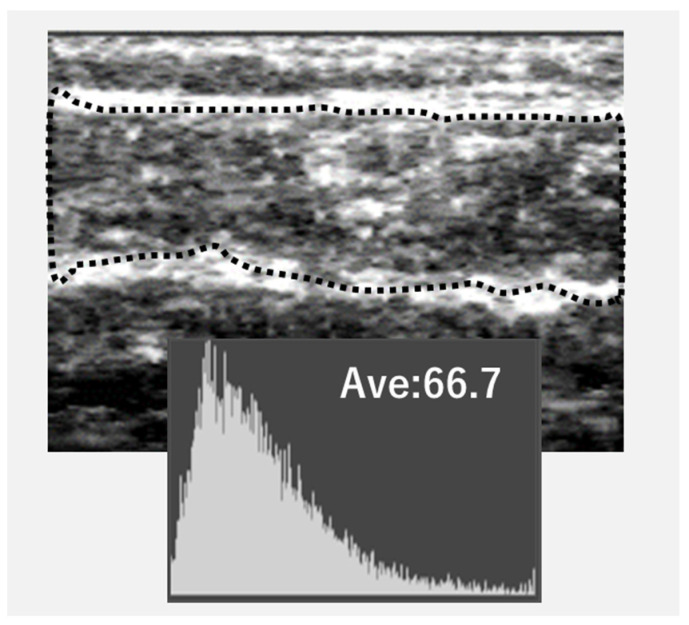
The analysis for echo intensity (EI) from the image.

**Figure 3 medicina-59-01894-f003:**
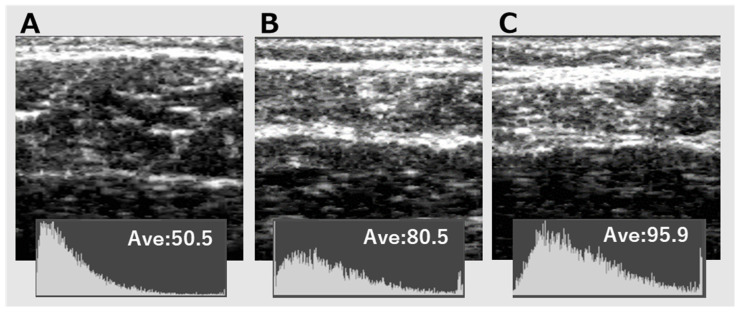
(**A**–**C**) Gastrocnemius images of a healthy adult and a patient with PAD.

**Table 1 medicina-59-01894-t001:** Characteristics of healthy adults and patients with PAD.

	Healthy Adults *n* = 73	PAD *n* = 35	T	χ^2^	*p* Value
Age; yr	73.2 ± 4.3	73.7 ± 8.5	0.45		0.65
BMI; kg/m^2^	23.2 ± 3.0	23.5 ± 3.2	0.33		0.74
BMI category; *n* (%) Normal range overweight obesity	56 (76.7) 17 (23.3) 0 (0)	23 (65.7) 12 (34.3) 0 (0)		2.1	0.341
History; *n* (%) Diabetes Hypertension Hyperlipidemia CKD CVD CVA Orthopedic diseases	- - - - - - -	21 (60.0) 31 (88.6) 24 (68.6) 6 (17.1) 5 (14.3) 5 (14.3) 8 (22.9)			
Bilateral lesions; *n*%	-	11 (31.4)			
SMI; kg/m²	8.0 ± 1.0	6.7 ± 0.8	−6.69		<0.01
Grip strength; kg	36.4 ± 6.0	30.1 ± 6.3	−5.06		<0.01
Walking speed; m/sec	1.9 ± 0.4	0.9 ± 0.2	−15.07		<0.01
6MD; m	-	344.8 ± 94.0			

Numerical data are expressed as mean ± SD in age, BMI, SMI, grip strength, walking speed, and 6MD. BMI, body mass index; CKD, chronic kidney disease; CVD, cardiovascular disease; CVA, cerebral vascular disease; SMI, skeletal muscle mass index; 6MD, 6 min walking distance.

**Table 2 medicina-59-01894-t002:** Comparisons of ultrasonography between the legs of healthy adults and legs with higher and lower ABI of patients with PAD.

	The Leg of Healthy Adults *n* = 73	The Legs of Patients with PAD		*T*	*U*
		Higher ABI *n* = 35	Lower ABI *n* = 35		
ABI	-	0.89 ± 0.16	0.62 ± 0.13	7.48	
Ultrasonography					
SFT; mm	2.3 (0.3–9.0)	3.1 (0.5–7.7)	3.0 (0.7–6.5)		6.1
GT; mm	13.3 (8.1–19.5)	11.3 (7.8–15.9) **	10.7 (7.8–18.1) **		32.8
EI; AU	72.0 (37.0–134.7)	80.8 (45.7–140.0)	83.6 (43.1–147.0) *		7.5

Numerical data are expressed as mean ± SD in ABI and as median (min–max) in SFT, GT, EI. ABI, ankle brachial pressure index; AU, arbitrary unit; EI, echo intensity; GT, gastrocnemius thickness; SFT, subcutaneous fat thickness. *, Significantly compared with the leg of healthy adults, *p* < 0.05; **, Significantly compared with the leg of healthy adults, *p* < 0.01.

**Table 3 medicina-59-01894-t003:** Associations of 6 min distance with gastrocnemius and echo intensity adjusted by age and body mass index via multiple regression analysis.

	Model 1		Model 2		Model 3		Model 4	
	*β*	*p* Value	*β*	*p* Value	*β*	*p* Value	*β*	*p* Value
Age	−0.25	0.35	−0.14	0.38	−0.89	0.38	−0.13	0.39
BMI	−0.17	0.15	−0.19	0.21	−1.33	0.19	−0.20	0.19
GT with higher ABI	0.86	0.64						
EI with higher ABI			−0.46	<0.01				
GT with lower ABI					0.26	0.19		
EI with lower ABI							−0.45	<0.01
*R*	0.29		0.54		0.36		0.52	
Adjusted *R*²	0.01		0.23		0.05		0.19	
*F*	0.37		4.49 *		1.63		3.97 *	

BMI, body mass index; EI, echo intensity; GT, gastrocnemius thickness. *, Significant value, *p* < 0.01.

## Data Availability

The research data in this study is unavailable due to ethical restrictions.

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
