# Peer review of "Echo Intensity of Gastrocnemius Is Independently Associated with 6-Minute Walking Distance in Male Patients with Peripheral Arterial Disease"

_medicina, 2023, doi:10.3390/medicina59111894_

Round 1

Reviewer 1 Report

Dear Authors,

You raised an interesting topic of relationship between muscle echo-intensity and the 6MWT results.

Unfortunatelly, there are some major concerns about the methodolody and obtained results:

1) Were there any power and sample size calculations performed priot to the begining of the study? What were the participants groups count based on? In methodlogy section, you mention 210 healthy adults, but in the analysis we can see 73 healty participants, could you explain the difference?

2) There is no such thing as "USG defult setting". You need to give all the parameters depth, focus, gain, frequency etc. 

3) You mentioned the intra-class correlation but as far as you have 2 examiners also inter-class correlation should be asessed. 

4) Table 3 is unclear to me. What are those "models"? I do not see there analysis between 6MWT and EI but the ABI results.

To review the discussion I need response to those issues.

Kind regards

Native speaker check-up suggested. 

Author Response

Dear reviewer 1

Thank you very much for your peer review and constructive comments in your busy schedule. I revised and added sentences, and underlined the revised parts and sentences.

I hope that my revision article will match your advice and comments.

Dear reviewer 1

1) Were there any power and sample size calculations performed priot to the begining of the study? What were the participants groups count based on? In methodlogy section, you mention 210 healthy adults, but in the analysis we can see 73 healty participants, could you explain the difference?

→ Thank you very much for your comments. I calculated the sample size for un-paired t test and X2 test in this study before the beginning of this study, and regarding multiple regression analysis, I calculated the power of analysis. Thus, I added the sentence about the sample size in statistical analysis section, page 4, line 182-183 as follows:

Moderate sample sizes were also calculated. The sample size was 14 participants in each group for un-paired t-test and X² test (tails, two; α-err, 0.05; power; 0.8).

And about the power of multiple regression analysis in results section, page 6, line 234-235 as follows:

The powers(1-β) of the multiple regression analysis for 35 participants were 0.81 in model 2 and 0.77 in model 4 which were high values.

Regarding inclusion criteria for 210 healthy adults, they comprised 77 males and 133 females, that was the reason why there was the difference. Therefore, I added sentences about inclusion criteria in more details in 2.1 participants section, page 2, line 75-86 as follows.

A total of 210 healthy adults aged mean 74.2±4.3 years, …with a mean BMI of 23.2±3.0 kg/m2 which was categorized into normal range 56(74.0%), and over-weight 17(23.3%) according to the WHO guideline (11).

2) There is no such thing as "USG defult setting". You need to give all the parameters depth, focus, gain, frequency etc.

→ Thank you very much for your comments. I agree with your comments, but ultrasound settings such as depth, focus, gain, and so on in this ultrasound device cannot be disclosed because of the patent in SAKAI Medical Science Co., Ltd, therefore, that would become limitation. Meanwhile, because the ultrasound settings in of this device have been locked and cannot be operated, fixed settings were used in this study. Therefore, I added the sentence about ultrasound settings in limitation section, page 8, line 322-325 as follows;

Fourth, the settings of this ultrasound device were not disclosed because it was patented by SAKAI Medical Science Co., Ltd, the low EI values obtained in this study could vary when using other ultrasound devices.

3) You mentioned the intra-class correlation but as far as you have 2 examiners also inter-class correlation should be assessed.

→ Thank you very much for your comments. I am sorry for giving confusing sentences. Examiner A measured only PAD patients and Examiner B measured only healthy adults, therefore, the inter-class correlation was not performed, and I revised the sentence of page 3, line119-121 as follows;

Ultrasonography was assessed in patients with PAD by examiner A and healthy adults by examiner B, who were specialists and well-trained examiners.

4) Table 3 is unclear to me. What are those "models"? I do not see there analysis between 6MWT and EI but the ABI results.

→ Thank you very much for your comments. I am sorry not to provide enough information for the Table 3.

Model 1 means that the multiple regression analysis that included 6MD as a dependent variable, GT with the higher ABI leg, age, and BMI as independent variables, and likewise, model 2 means EI with the higher ABI leg, age, and BMI as independent variables, and model 3 means GT with the lower ABI leg, age, BMI as independent variables, and model 4 means EI with the lower ABI leg, age, and BMI as independent variables. Therefore, I added above-mentioned sentences in Page 6, line 227-231 as follows:

Table 3 shows that the multiple regression analysis that…and EI with the lower ABI leg, age, and BMI as independent variables in model 4.

Regarding association of 6MD with the ABI, lots of previous studies have been clarifying that there was no association walking distance and ABI. And, effects of higher and lower ABI on 6MD could be included in each of GT and EI with the higher and lower ABI. Because of these reasons, we didn’t include the ABI considering over fitting for analysis.

Sincerely yours,

Satoshi Yuguchi

Reviewer 2 Report

Dear authors,

Manuscript ID: medicina-2631499

Title Manuscript: Echo Intensity of Gastrocnemius is Independently Associated with 6-minute Walking Distance in Male Patients with Peripheral Arterial Disease

This is an interesting and important topic since the study participants are patients with peripheral arterial disease MAJOR REVISIONS are necessary in order to make it suitable for a final decision for “Medicina”;

Summary statement: This study examined the association between a skeletal muscle in the lower limbs and the 6-min walking distance (6-MD) in patients with peripheral arterial disease (PAD) and as well as association of the thickness and echo intensity (EI) of gastrocnemius measured by ultrasonography with 6MD in patients with PAD. The gastrocnemius thickness (GT) and EI were measured using an ultrasound device in 35 male PAD patients and 73 age- and gender-matched healthy adults. The results showed that GT was considerably greater in healthy adults than in both legs with higher and lower ankle brachial pressure index (ABI) (median values, 13.3 vs. 11.3 vs. 10.7, P < 0.01), whereas EI was lower in healthy adults than in the lower ABI leg (72.0 vs. 80.8 vs. 83.6, P < 0.05). 6-MD was shown to be substantially related to EI in both legs with higher and lower ABI (P < 0.01) but not to GT. In PAD, GT was lower, and EI was higher than in healthy adults.

POINTs of STRENGTH:

1) Association of thickness and echo intensity (EI) of gastrocnemius with 6-min walking distance (6-MD) in patients with peripheral arterial disease; 

POINTs of WEAKNESS (and/or should be revised to improve the manuscript):

Abstract

2) Please add the type of study, mean weight and BMI for participants in the “methods” section of the abstract;

3) The variable of 6-min walking distance in the methods section of the abstract is unclear. Please provide;  

1. Introduction

4) The hypothesis and purposes of this study are not specified. Please specify;

2. Materials and Methods

5) Please remove this wrong and irrelevant content from the “Materials and methods” section [line N. 62-76]; before submitting the article, authors should review their manuscript several times to avoid such common mistakes:

“[The Materials and Methods should be described with sufficient details to allow oth-62 ers to replicate and build on the published results. Please note that the publication of your 63 manuscript implicates that you must make all materials, data, computer code, and proto-64 cols associated with the publication available to readers. Please disclose at the submission 65 stage any restrictions on the availability of materials or information. New methods and 66 protocols should be described in detail while well-established methods can be briefly de-67 scribed and appropriately cited. 68

Research manuscripts reporting large datasets that are deposited in a publicly avail-69 able database should specify where the data have been deposited and provide the relevant 70 accession numbers. If the accession numbers have not yet been obtained at the time of 71 submission, please state that they will be provided during review. They must be provided 72 prior to publication. 73

Interventionary studies involving animals or humans, and other studies that require 74 ethical approval, must list the authority that provided approval and the corresponding 75 ethical approval code.].”

2.1. Participants

6) The recruitment process and/or screening of study participants, especially inclusion and exclusion criteria should be described in more detail such as initial sample size, gender, age range, BMI category based on the WHO, blood pressure, drug interventions and or free of medications, physical fitness level, healthy status, and so on.

2.2. Study Design

7) Please specify measurement methods of variables in the “study design” section;

2.3. Procedures

2.3.1 Ultrasonography

8) Was the ultrasonography measurement evaluated by a specialist? IF YES, please specify. Furthermore, were all participants measured by the same specialist OR expert? Please specify;

2.4. Statistical Analysis

9) Did authors use a statistical software to calculate the sample size? If YES, please explain and add its name and results in the “Statistical Analysis” section;

10) The significance level of statistical analysis was considered for one-tailed OR two-tailed? Please clarify;

3. Results

11) Results section is well written;

4. Discussion & 5. Conclusion

12) What does this study add to the literature? Please explain and add in the conclusions section;

13) Please provide clinical implications for patients with peripheral arterial disease at the end of this manuscript;

Best Regards

 8 October 2023

Author Response

Dear reviewer 2

Thank you very much for your peer review and constructive comments in your busy schedule. I revised and added sentences, and underlined the revised parts and sentences.

I hope that my revision article will match your advice and comments.

1) Please add the type of study, mean weight and BMI for participants in the “methods” section of the abstract;

→ Thank you very much for your comments. In abstract, line 16-18, I added the sentence, the type of study, mean age and BMI in participants as follows;

This cross-sectional study included 35 male patients PAD (mean age, 73.7 years; mean body mass index [BMI], 23.5 kg/m2) and age- and gender-matched healthy adults (mean age, 73.2 years; mean BMI, 23.3 kg/m2).

2) The variable of 6-min walking distance in the methods section of the abstract is unclear. Please provide; 

→ I apologize for confusing sentences. I added and revised sentences in line 19- 22 as follows;

Both legs of patients with PAD were classified based on higher and lower ankle brachial pressure index (ABI), and the GTs and EIs with higher and lower ABI were compared with those of healthy adults. Multiple regression analysis, incorporated 6MD as a dependent variable and each of GT and EI with higher and lower ABI, age, and BMI as independent variables.

Introduction

3) The hypothesis and purposes of this study are not specified. Please specify;

→Thank you very much for your comments. I added sentences of the hypothesis and purposes of this study in details in page 2, line 45-46, line 59-65, and line 66-69 as follows;

Line 45-46: However, the relationship between the abnormalities and walking distance in patients with PAD presenting IC have not been studied sufficiently.

Line 59-65: Moreover, it has been unclear whether there were …after revascularization and exercise therapy, in patients with PAD.

Line 66-69: Thus, the purpose of this study was to clarify the following points: to examine the differences in gastrocnemius thickness (GT) and EI based on ultrasonography between healthy adults and patients with PAD presenting IC and to determine the associations of GT and EI with 6-min walking distance (6MD) in patients with PAD.

Materials and Methods

4) Please remove this wrong and irrelevant content from the “Materials and methods” section [line N. 62-76]; before submitting the article, authors should review their manuscript several times to avoid such common mistakes:

→ I am very sorry for my mistakes. I removed the sentences.

Participants

5) The recruitment process and/or screening of study participants, especially inclusion and exclusion criteria should be described in more detail such as initial sample size, gender, age range, BMI category based on the WHO, blood pressure, drug interventions and or free of medications, physical fitness level, healthy status, and so on.

→Thank you very much for your comments. We collected only data of presence or absence of drug intervention and medication, using any walking aid, and independence of activities of daily living, which would be limitations in this study. However, it is very important points, so I added the following sentence in limitation section, page 8, line 319 to 321 as follows;

Second, because health status data, including blood pressure, activity level, and the type of drugs were not collected in healthy adults, the detailed state of participants was unclear.

Meanwhile, I added the inclusion criteria for participants in more details, such as mean BMI and BMI category based on the WHO guideline which were added sentences in Table 2 and in page 2, line 75-86 as follows:

A total of 210 healthy adults aged mean 74.2±4.3 years, …with a mean BMI of 23.2±3.0 kg/m2 which was categorized into normal range 56(74.0%), and over-weight 17(23.3%) according to the WHO guideline (11).

Study Design

6) Please specify measurement methods of variables in the “study design” section;

→I am sorry to have written hard-to-understand sentences. I revised section forms and specified measurement methods in study design section.

Ultrasonography

7) Was the ultrasonography measurement evaluated by a specialist? IF YES, please specify. Furthermore, were all participants measured by the same specialist OR expert? Please specify;

→Thank you very much for your comments. Examiner A measured PAD patients and examiner B measured healthy adults, therefore, two examiners measured each of data in PAD patients and healthy adults. And two examiners were specialists and well-trained examiners. I added the sentences as follows in page 3, line 119-121 as follows;

Ultrasonography was assessed in patients with PAD by examiner A and healthy adults by examiner B, who were specialists and well-trained examiners. 

Statistical Analysis

8) Did authors use a statistical software to calculate the sample size? If YES, please explain and add its name and results in the “Statistical Analysis” section;

→ Thank you very much for your comments. I calculated the sample size about un-paired t-test and X2 test before analyses by using SPSS ver27(IBM Corp.). I added the sentence about analysis of the sample size in statistical analysis section, page 4, line 182 to 183 as follows:

Moderate sample sizes were also calculated. The sample size was 14 participants in each group for un-paired t-test and X² test (tails, two; α-err, 0.05; power; 0.8).

Regarding multiple regression analysis, I calculated powers of analyses based on 35 participants after analysis. I added the sentence about the powers of analysis in results section, page 6, line 234-235 as follows:

The powers(1-β) of the multiple regression analysis for 35 participants were 0.81 in model 2 and 0.77 in model 4 which were high values.

9) The significance level of statistical analysis was considered for one-tailed OR two-tailed? Please clarify;

→ Thank you very much for your comments. All of significant level of statistical analysis were two-tailed. I added the sentence in statistical analysis section, page 4, line 190 as follows;

 a significant level of < 0.05(two-tailed) was set.

Discussion & 5. Conclusion

10) What does this study add to the literature? Please explain and add in the conclusions section;

11) Please provide clinical implications for patients with peripheral arterial disease at the end of this manuscript;

→ Thank you very much for your comments. I revised sentences and added important points in this study, perspectives, and clinical implications in this study in page 7, line 252-254, and page 8, line 305-313 in discussion section, and line 329-331 and 333-336 in conclusion section as follows:

・Page 7, line 252-254

This is the first report that utilizes an ultrasound device to detect skeletal muscle abnor-malities in patients with PAD, and clarify the relationships between EIs of the gas-trocnemius in both of the legs and 6MD.

・Page 8, line 305-313

Important points of this study were that ultrasonography…or occurrence of any adverse events, should be determined.

・Page 8, line 329-331 and 333-336

Because patients with PAD had…intramuscular fat in patients with PAD.

This study showed that the EI obtained by ultrasonography・・・・ for male patients with PAD presenting IC.

Sincerely yours,

Satoshi Yuguchi

Round 2

Reviewer 1 Report

Dear Authors,

Thank you for the responces. Most of them are clear and I accept your replays. Unfortunately, I cannot accept the explanation that USG parameters are confidential due to the copyrights of the device manufacturer. Every ultrasound displays the depth of focus, the gain etc as those parameters are not fixed and need to be adjusted to every patient and every examination. I once again ask you to provide the information on what were "the default settings".

Kind regards

Author Response

Dear Reviewer 1

Thank you very much for your peer reviewing and providing important comments. I revised and added sentences, and underlined the revised parts and sentences.

I hope that my answer and explanation will resolve your concerns.

Thank you for the responces. Most of them are clear and I accept your replays. Unfortunately, I cannot accept the explanation that USG parameters are confidential due to the copyrights of the device manufacturer. Every ultrasound displays the depth of focus, the gain etc as those parameters are not fixed and need to be adjusted to every patient and every examination. I once again ask you to provide the information on what were "the default settings".

→Thank you for your comments and concerns. As the reviewer mentioned, when measuring ultrasound for some targets, it is needed to adjust various settings for each examination. However, the echo intensity (EI) of skeletal muscles is influenced by settings such as gain and depth of focus. Consequently, adjusting these settings individually for each case causes the loss of reliability and validity for echo intensity (EI) measurements. Because of those characteristics, when only measuring EI, it is essential to use fixed settings such as gain, depth of focus, and so on, and many previous studies used fixed values for gain, depth of focus, and other parameters without individual adjustments for each case, and also utilized the same equipment for their investigations.

“Default settings” in this study means settings that the gain, depth of focus, and other parameters were preconfigured for the skeletal muscle and designed to be non-modifiable by the manufacturer. In addition, similar to previous studies, all subjects were measured under the same conditions in this study. However, “default settings” could be confusing words. Thus, I revised the sentence, page 3, line 102-106 as follows;

The measurement of ultrasonography for all participants was conducted using fixed settings, such as B-mode, a fixed dB dynamic range, a fixed gain, and a fixed depth of focus, that were preconfigured for the skeletal muscle and designed to be non-modifiable by the manufacturer.

Sincerely yours,

Satoshi Yuguchi

Reviewer 2 Report

Dear Authors,

Manuscript ID: medicina-2631499

Title Manuscript: Echo Intensity of Gastrocnemius is Independently Associated with 6-minute Walking Distance in Male Patients with Peripheral Arterial Disease

In general, although this manuscript has found good content after correcting major revisions and these revisions are modified and added in this manuscript, one concern OR MINOR REVISION has to be addressed before a final version can be made:

Abstract

Materials and Methods:

1) The authors reported for healthy participants [age- and gender-matched healthy adults] in the abstract, whereas in the “2. Materials and Methods” section for participants has been reported this sentence “A total of 210 healthy adults aged mean 74.2±4.3 years, comprising 77 males and 133 females”. Which is true? Were both genders considered? OR only males? Please clarify and correct;

Best Regards

16 October 2023

Author Response

Dear Reviewer2

Thank you very much for your peer reviewing and providing important comments. I revised and added sentences, and underlined the revised parts and sentences.

I hope that my answer and explanation will resolve your concerns.

In general, although this manuscript has found good content after correcting major revisions and these revisions are modified and added in this manuscript, one concern OR MINOR REVISION has to be addressed before a final version can be made:

Abstract

Materials and Methods:

1) The authors reported for healthy participants [age- and gender-matched healthy adults] in the abstract, whereas in the “2. Materials and Methods” section for participants has been reported this sentence “A total of 210 healthy adults aged mean 74.2±4.3 years, comprising 77 males and 133 females”. Which is true? Were both genders considered? OR only males? Please clarify and correct;

→Thank you very much for your comments, and I am sorry to confusing you because of my insufficient explanation.

This study targets 73 male healthy adults (mean age, 73.2 years; mean BMI, 23.3 kg/m2) who met the inclusion criteria out of a total 210 male and female healthy adults aged mean 74.2±4.3 years, comprising 77 males and 133 females. Thus, I added the sentence in abstract, page 1, line 18 as follows;

age- and gender-matched 73 male healthy adults (mean age, 73.2 years; mean BMI, 23.3 kg/m2).

In addition, I added the sentence in materials and Methods section, page 2, line 83-84 as follows;

finally, 73 male healthy adults were targeted in this study, who were 73.2±4.3 years old, with a mean BMI of 23.2±3.0 kg/m2

Sincerely yours,

Satoshi Yuguchi